# Rational construction of a reversible arylazo-based NIR probe for cycling hypoxia imaging in vivo

Yuming Zhang[1,2], Wenxuan Zhao [1], Yuncong Chen [1 ✉], Hao Yuan[1], Hongbao Fang[1], Shankun Yao[1], Changli Zhang[3], Hongxia Xu[1], Nan Li[4], Zhipeng Liu[5], Zijian Guo [1 ✉], Qingshun Zhao [4 ✉], Yong Liang [1] & Weijiang He [1 ✉]

Reversible NIR luminescent probes with negligible photocytotoxicity are required for long-term tracking of cycling hypoxia in vivo. However, almost all of the reported organic fluorescent hypoxia probes reported until now were irreversible. Here we report a reversible arylazo-conjugated fluorescent probe (HDSF) for cycling hypoxia imaging. HDSF displays an off-on fluorescence switch at 705 nm in normoxia-hypoxia cycles. Mass spectroscopic and theoretical studies confirm that the reversible sensing behavior is attributed to the two electron-withdrawing trifluoromethyl groups, which stabilizes the reduction intermediate phenylhydrazine and blocks the further reductive decomposition. Cycling hypoxia monitoring in cells and zebrafish embryos is realized by HDSF using confocal imaging. Moreover, hypoxic solid tumors are visualized and the ischemia-reperfusion process in mice is monitored in real-time. This work provides an effective strategy to construct organic fluorescent probes for cycling hypoxia imaging and paves the way for the study of cycling hypoxia biology.

[1] State Key Laboratory of Coordination Chemistry, School of Chemistry and Chemical Engineering, Chemistry and Biomedicine Innovation Center (ChemBIC), Nanjing University, Nanjing, China. [2] School of Chemistry and Chemical Engineering, Nantong University, Nantong, China. [3] School of Environmental Science, Nanjing Xiaozhuang University, Nanjing, China. [4] Model Animal Research Center, Nanjing University, Nanjing, China. [5] College of Materials Science and Engineering, Nanjing Forestry University, Nanjing, China. ✉email: chenyc@nju.edu.cn; zguo@nju.edu.cn; qingshun@nju.edu.cn; heweij69@nju.edu.cn

ycling hypoxia, characterized by spatial and temporal $O_2$ fluctuations associated with hypoxia-reoxygenation, has attracted increasing attention these days, since it is a key feature of the tumor microenvironment, and is involved in ischemia-reperfusion injuries (IRI) of variable tissues and organs[1–7]. It could upregulate the hypoxia-inducible factor 1α (HIF 1α) for anaerobic metabolism[8–10], and contribute more to tumor angiogenesis, metastasis, and drug resistance when compared with chronic hypoxia[11–13]. Such a hypoxia-reoxygenation process could generate reactive oxygen species (ROS) in mitochondria and cause IRI in heart attack and stroke[14]. As cycling hypoxia is mainly a phenomenon lasting for a long period in tissues and organs, long-term tracking of cycling hypoxia in live animal models is highly demanded by both pathological and clinical studies. With the advantages including rapid response and high sensitivity/resolution, optical imaging is more effective for cycling hypoxia tracking than positron emission tomography imaging[15,16], immunostaining[17–19], and magnetic resonance imaging[20–22]. However, optical imaging for cycling hypoxia in mammalian animal models such as mice has not been realized so far, and the reversible hypoxia luminescent probes with near-infrared (NIR) emission is especially appreciated.

Besides phosphorescent metal complex-based probes for cycling hypoxia sensing[23–28], organic fluorescent probes are always promising candidates due to their lower photocytotoxicity in long-term imaging than phosphorescent metal complexes. To date, almost all available organic fluorescent probes for hypoxia are reducible compounds containing nitrobenzenes[29–31], quinones[32], N-oxides[33], and arylazos[34–40]. However, the hypoxia-induced reduction sensing mechanisms of these reported probes lead to irreversible sensing behavior (Fig. 1a), which could give false information when the "turn-on" fluorescent molecules translocate to the normoxia region and not suitable for cycling hypoxia tracking. Exploring a reliable strategy to design organic fluorescent probes showing reversible hypoxia response is in urgent need yet quite challenging[41,42].

Herein, we report a reversible fluorescent probe for cycling hypoxia, HDSF (Fig. 1, and Supplementary Fig. 1), which was constructed by coupling 3,5-ditrifluoromethylbenzene with a NIR emissive xanthene/cyanine fused fluorophore (HD) via an azo-linker[43]. Its analog with a 4-(N,N-dimethyl)aminoazobenzene group, HDMA, was also prepared for comparison[34]. Besides its mitochondria targeting ability, HDSF displayed a reversible turn-on NIR fluorescent response to hypoxia. Experimental test and theoretical simulation suggested that the reversible sensing behavior was attributed to the stabilization of arylazo reduction intermediate phenylhydrazine by the electron-withdrawing tri-fluoromethyl groups, which blocked the subsequent reductive cleavage of N–N bond. Besides intracellular cycling hypoxia imaging, HDSF was able to monitor drug-induced hypoxia-reoxygenation cycles in zebrafish embryos via confocal imaging. Profiting from the NIR emission of HDSF, optical imaging for hypoxia in solid tumors, and real-time visualization of the ischemia-reperfusion process were realized in mice.

## Results

**Design and synthesis of HDSF.** Arylazo reduction by azo-eductase with dihydronicotinamide adenine dinucleotide phosphate (NADPH) as a cofactor in hypoxic microenvironments has been successfully utilized to construct hypoxia probes via integrating an arylazo with a fluorophore using azo as a linker (Fig. 1a)[34–40]. Arylazos are efficient emission quenchers due to their fast photo-induced E-Z isomerization[40,44,45], which makes these arylazo-conjugated probes almost non-fluorescent. The turn-on response to hypoxia was realized by reductive

decomposition of azo-linker to form aniline and the emissive amino-substituted fluorophore through intermediates arylazo radical and/or phenylhydrazine[40,46–48]. The reductive decomposition of azo-linker results in the irreversible sensing behavior for hypoxia, which makes these arylazo-conjugated probes incapable of sensing cycling hypoxia. It is supposed that modifying arylazo for reversible reduction might avert reductive decomposition to realize reversible hypoxia sensing.

Since the free energy elevation of the intermediates by electron-donating group (EDG) dimethylamine favors this irreversible reduction cleavage (vide infra)[46], we envisioned that the free energies of azo anion radical or phenylhydrazine intermediates might be significantly decreased by modifying arylazo moiety with electron-withdrawing groups (EWGs). This could stabilize the intermediate and enlarge the activation energy ($E$a) needed for subsequent reduction. Therefore the irreversible reductive cleavage of hydrazine could be prevented and it would result in a reversible response to hypoxia, providing that phenylhydrazine is $O_2$ sensitive.

As hypoxia induces mitochondria autophagy and biogenesis[49–51], the mitochondria-targeting hypoxia probes are appreciated for the additional advantage to clarify the relationship between mitochondria and hypoxia. In HDSF, a xanthene/cyanine fused fluorophore (HD) was adopted due to its lipophilic cation feature and NIR emission, which could be ideal for potential mitochondria targeting and in vivo imaging. Since the electron-withdrawing nitro group could be converted into an electron-donating amino group in the presence of nitroreductase in hypoxic microenvironments, the redox inert trifluoromethyl (-$CF_3$) group was selected as the EWG. HDSF was easily prepared by a two-step procedure starting from 3,5-bis(trifluoromethyl) aniline, and HDMA with an EDG was synthesized according to a reported procedure as a control probe. Detailed synthetic procedures are presented in supporting information. Both compounds were fully characterized by $^1$H nuclear magnetic resonance ($^1$H NMR), $^{13}$C nuclear magnetic resonance ($^{13}$C NMR), and high-resolution mass spectrum (HR-MS).

**Fluorescent sensing behavior of HDSF to hypoxia.** The cycling hypoxia sensing response of HDSF and HDMA were investigated by a standard procedure in the presence of rat liver microsomes (RLM) containing various reductases, and NADPH as electron donor[35,39]. From normoxia to hypoxia, absorption of HDMA and HDSF all displayed a red shift from 650 to 700 nm, respectively (Supplementary Fig. 8). HDSF solution in normoxia condition showed a weak broad NIR emission band (660–750 nm; $\lambda_{ex}$, 650 nm; $\Phi = 0.9\%$; $\varepsilon = 3.2 \times 10^4\,M^{-1}\,cm^{-1}$), and a distinct emission enhancement was recorded (~6-fold; $\lambda_{em}$, 705 nm; $\Phi = 7.5\%$; $\varepsilon = 7.6 \times 10^4\,M^{-1}\,cm^{-1}$) under hypoxia within 15 min (Fig. 1c and Supplementary Fig. 9). In contrast, HDMA showed only a slight emission enhancement from normoxia to hypoxia in the same experimental conditions (Fig. 1e). When the solutions were set back to normoxia, the fluorescence of HDSF decreased back to the original level while that of HDMA showed no obvious change. The "on-off" response of HDSF in hypoxia-normoxia treatment could operate at least for three cycles with negligible intensity loss (Fig. 1c, d). The color of HDSF solution in the presence of RLM and NADPH changed from blue in normoxia to blue-green in hypoxia and then back to blue in normoxia (Supplementary Fig. 11). All these results clearly demonstrated that HDSF was able to sense hypoxia in a reversible manner, while HDMA only showed the irreversible response.

The fluorescent sensing selectivity of HDSF for hypoxia over other biorelated chemical species was also investigated (Supplementary Fig. 12). The addition of 2 mM $Na^+$, $K^+$, $Ca^{2+}$, $Mg^{2+}$

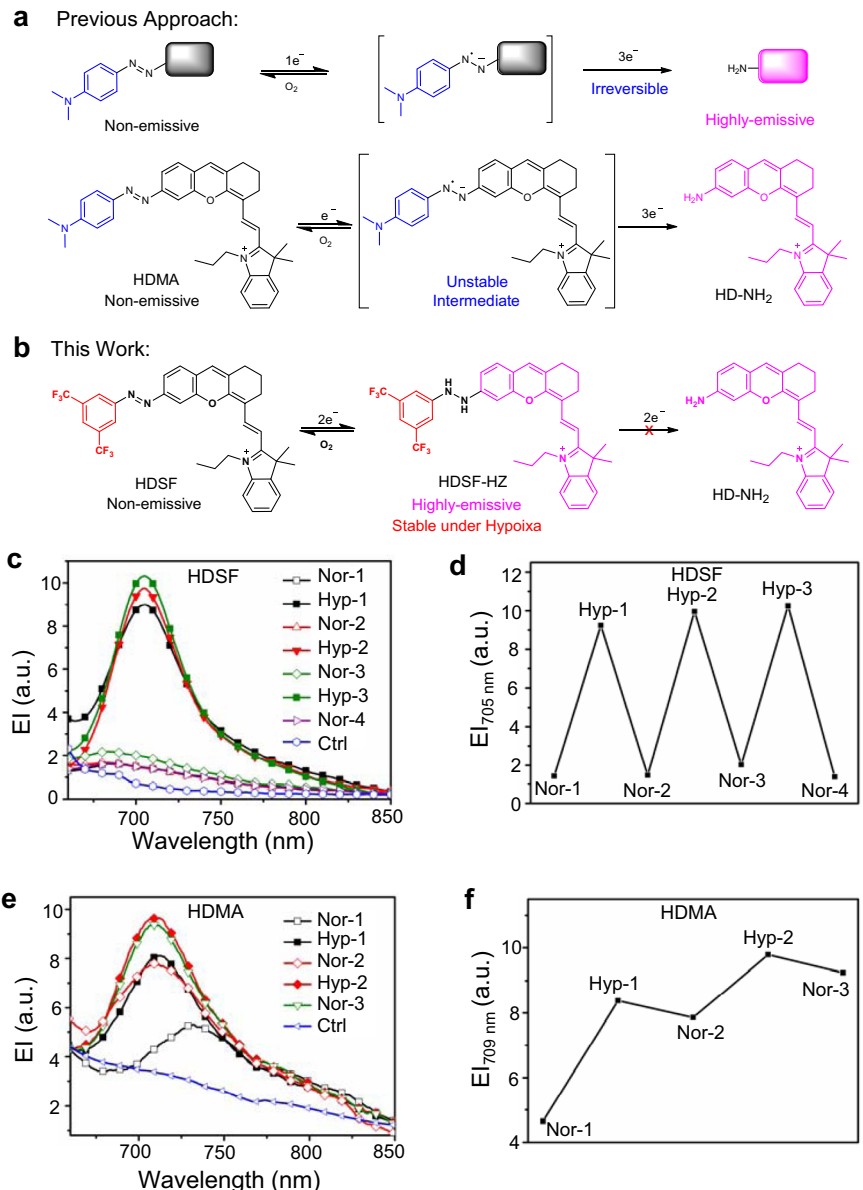

**Fig. 1 Reversible imaging mechanism and behavior of HDSF. a** Normal azobenzene-derived fluorescent probes for hypoxia and their reductive decomposition of N–N bond by the hypoxic microenvironment in living systems. **b** The reversible hypoxia sensing mechanism of probe HDSF. The square and fluorophore in black represent the quenched-fluorophore, and those in bright magenta represent the emitting-fluorophore. **c** Fluorescent spectra of 20 µM HDSF in PBS buffer (0.1 M, pH 7.4, 2% DMSO, v/v) containing rat liver microsomes (RLM, 250 µg mL$^{-1}$) and NADPH (100 µM) recorded in normoxia-hypoxia cycles. **d** Fluorescence intensity of HDSF at 705 nm detected in (**c**). **e** Fluorescent spectra of 20 µM HDMA in PBS buffer (0.1 M, pH 7.4, 2% DMSO, v/v) containing RLM (250 µg mL$^{-1}$) and NADPH (100 µM) recorded in normoxia-hypoxia cycles. **f** Fluorescence intensity of HDMA at 709 nm detected in (**e**). Ctrl: PBS buffer with RLM and NADPH. λ$_{ex}$, 650 nm.

and 50 µM $Mn^{2+}$, $Fe^{2+}$, $Fe^{3+}$, $Co^{2+}$, $Ni^{2+}$, $Cu^{2+}$, $Zn^{2+}$ did not induce substantial fluorescence change of HDSF, and other common species in living systems, including reactive nitrogen, oxygen, and sulfur species (RNS, ROS, and RSS), reductive ascorbic acid and oxalic acid, showed no obvious influence on HDSF emission. Only hypoxic condition could trigger such a distinct emission enhancement of HDSF. Meanwhile, the fluorescence of HDSF exhibited no obvious pH-dependence from pH 3.0 to pH 9.0 (Supplementary Fig. 13). All these results indicated that HDSF could be an ideal candidate for cycling hypoxia imaging in living systems.

**Hypoxia sensing mechanism of HDSF**. To clarify this hypoxia sensing behavior, electron paramagnetic resonance (EPR) assay

was conducted first, and no obvious free radical signal was found in hypoxic HDSF solution containing RLM and NADPH, with or without the radical capture agent 5,5-Dimethyl-1-pyrroline-N-oxide (DMPO, Supplementary Fig. 14). This data demonstrated that the single electron reduction product azo anion radical of HDSF was not stable enough to be captured, suggesting a mechanism different from previous reported arylazo-reduction based hypoxia probes (Fig. 1a)[34–40]. Electrospray ionization mass spectrum (ESI-MS) determination of HDSF after hypoxia incubation with RLM and NADPH showed a new MS peak of m/z 638.42 (Fig. 2a), which could be assigned as the two-electron-reduction product phenylhydrazine derivative (HDSF-HZ). MS signal for the proposed reductive decomposition product HD-$NH_2$ (Fig. 1b) was not observed, suggesting that the cleavage of the

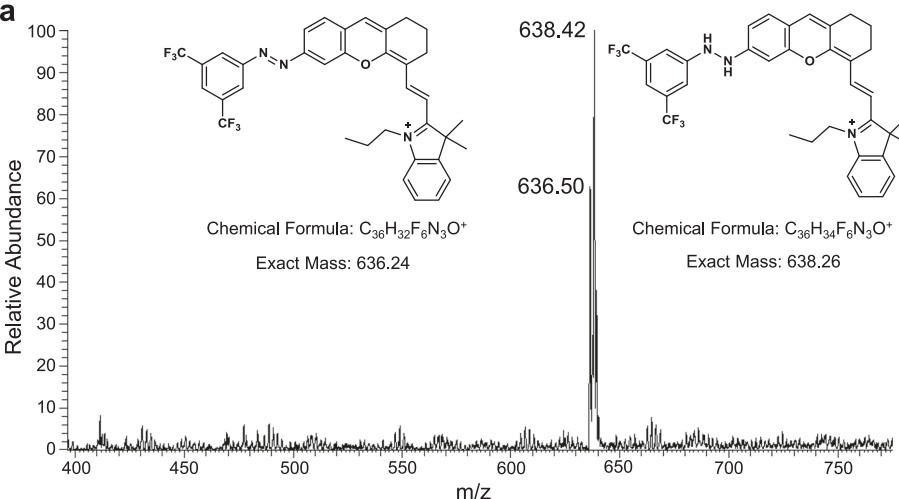

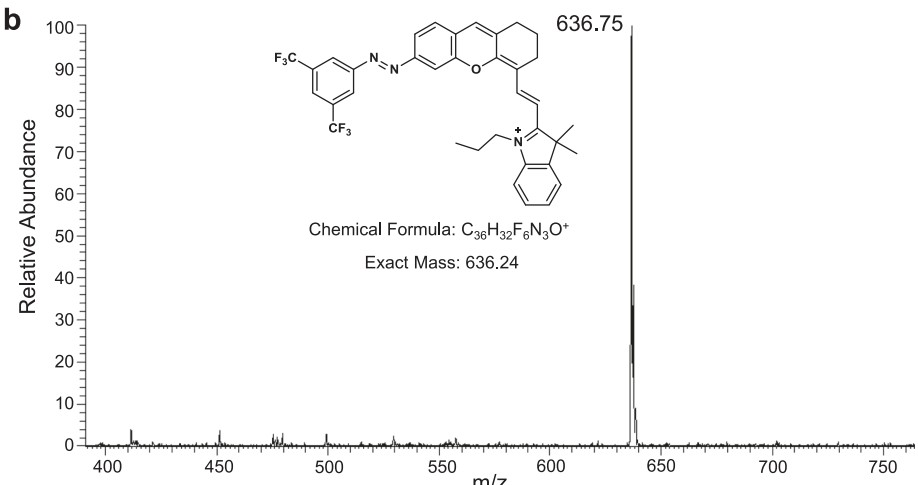

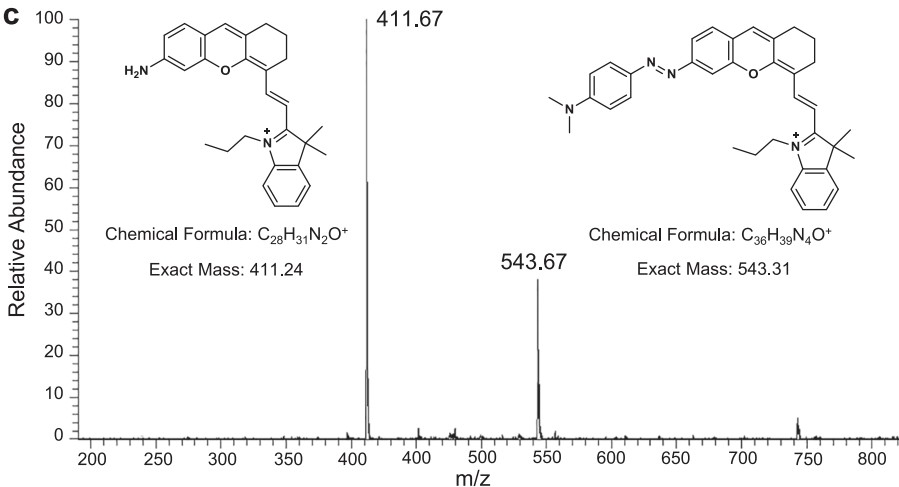

**Fig. 2 The reversible reaction of HDSF in hypoxia-reoxygenation cycle.** ESI-MS spectra of probe solutions (20 µM in PBS buffer) incubated with RLM (250 µg mL$^{-1}$) and NADPH (100 µM), **a** HDSF incubated in hypoxic condition. The supernatant of the reaction solution was directly used for detection. **b** HDSF solution exposed in the air after incubated in the hypoxic environment. **c** HDMA incubated in hypoxic condition. Solutions of (**b**) and (**c**) were quenched with MeCN, then vortexed and centrifuged, and the organic layers were used for detection respectively.

phenylhydrazine N–N bond was effectively blocked. After a hypoxia-normoxia cycle, the MS signal for HDSF-HZ disappeared and only the signal for probe HDSF (m/z of 636.75) was observed, confirming that arylazo group was recovered under normal $O_2$ content (Fig. 2b). However, hypoxia incubation of HDMA with RLM and NADPH resulted in a main MS signal of m/z 411.67 for HD-NH$_2$ and a minor signal for HDMA (Fig. 2c), suggesting most HDMA underwent azo reduction decomposition to form HD-NH$_2$ (Fig. 1a). The above data confirmed that HDSF reduction by RLM and NADPH in hypoxic condition could be stabilized in the form of phenylhydrazine, i.e., HDSF-HZ, which could recover to HDSF when the $O_2$ level was elevated (Fig. 1b)[47,48].

Density Functional Theory (DFT) calculation was also conducted to investigate the reversible reduction mechanism of HDSF. In order to mimic the physiological matrix, a simplified model of NADPH, reduced nicotinamide (1-methyl-1,4-dihydropyridine-3-carboxamide) was selected as the reductive agent, while $H_2PO_4^-$ was adopted as a proton donor for calculations. The DFT-computed free energies for intermediates and products in the reduction processes of HDSF and HDMA are shown in Fig. 3 (For computational details and references, see Supporting Information). In the case of HDSF, a stable deprotonated hydrazine intermediate is generated by hydride transfer from NADPH to HDSF. Several further reduction pathways have been considered. Among them, one-electron reduction followed by the N–N bond cleavage has the lowest barrier, which is $25.8\ kcal\ mol^{-1}$ (from Int1 to TS, Fig. 3). However, from a kinetic view, such a barrier is still very difficult to overcome at physiological temperature, making the reduction stop at the hydrazine intermediate. This is in accordance with the fluorescence change and ESI-Mass spectra result. By contrast, the first hydride reduction step for HDMA is endergonic by $7.1\ kcal\ mol^{-1}$, indicating that the hydrazine intermediate is unstable. Further one-electron reduction and fragmentation is facile, with an overall barrier of $22.4\ kcal\ mol^{-1}$ (from HDMA to TS, Fig. 3). This is $3.4\ kcal\ mol^{-1}$ lower than that for HDSF, corresponding to ~300 times increase in rate constant. Therefore, this transformation can smoothly occur at 37 °C. The formation of final products is exergonic by $13.5\ kcal\ mol^{-1}$, resulting in a $35.9\ kcal\ mol^{-1}$ barrier for the reverse reaction, which is impossible to overcome under mild conditions. Therefore, the reduction of HDMA irreversibly generates HD fluorophore, consistent with experimental findings (Figs. 1 and 2c). From these computational results, it is clear that the electron-withdrawing nature of trifluoromethyl groups stabilizes the electron-rich hydrazine intermediate but destabilizes the electron-poor nitrogen radical forming in the N–N bond cleavage transition state. This causes the hypoxia reduction of HDSF to

only generate a stable phenylhydrazine intermediate, which can be reversed to azobenzene by reoxygenation. Therefore, construction of fluorophore with EWG-substituted arylazo could serve as an effective strategy to design reversible hypoxia probes.

**Confocal imaging of cycling hypoxia in live cells**. Confocal fluorescence imaging was conducted in MCF-7 cells (Fig. 4). The intracellular fluorescence of HDSF-stained cells was negligible under normoxia (Fig. 4c). The fluorescence increased significantly with the decrement of $O_2$ content in the incubation environment. Subcellular co-localization of HDSF with Mito-Tracker Green and LysoSensor Green was investigated in a hypoxic condition (Fig. 4a, b). The fluorescence of HDSF overlapped well with that of Mito-Tracker Green with a Pearson's correlation coefficient of 0.92. However, the correlation coefficient of HDSF with LysoSensor Green was only 0.45. This mitochondria targetability of HDSF made it more attractive for the imaging application to understand hypoxia-associated mitochondria dysfunction. On the other hand, the bright intracellular fluorescence decreased distinctly when the incubation condition was switched from hypoxia to normoxia (Fig. 4c). The on-off fluorescence switch in HDSF-stained cells was also observed in the subsequent deoxygenation and reoxygenation process (Fig. 4d). Furthermore, cells co-stained with HDSF and a commercial irreversible hypoxia reagent Image-iT$^{TM}$ Green under deoxygenation and reoxygenation process (Supplementary Fig. 17). Fluorescence in HDSF channel repeated the on-off switch, while fluorescence in Image-iT$^{TM}$ Green channel turned on in hypoxia and didn't change with the environmental $O_2$ content. In addition, since arylazo is reduced by azoreductase with NADPH as an electron donor[35,39], substance that inhibits electron donor or depresses azoreductase activity could influence the arylazo reduction. Diphenyliodonium chloride (DPI) is an inhibitor of electron transporter and dicoumarol (DIC) is an inhibitor of azoreductases[52]. Cells pretreated with DIC and DPI respectively were all dimmed obviously compared with control (Supplementary Fig. 19). These results illustrated that the intracellular fluorescence increment under hypoxia was a result of HDSF reduction, and re-quenched when $O_2$ content was raised. Although there was almost no fluorescence in HDSF-stained cells under normoxia, distinct intracellular fluorescence could be observed under a moderate hypoxia condition of 10% $O_2$ (Fig. 4c), suggesting that HDSF was more sensitive to hypoxia than other reported arylazo-derived probes[35,39].

**Imaging of cycling hypoxia in zebrafish and mice**. The intracellular imaging performance of HDSF for cycling hypoxia

**Fig. 3 DFT-calculated free energies for the reduction processes of HDSF and HDMA.** Numbers in parentheses are in kcal/mol, which are free energies of HDSF reduction intermediates and products (blue color) or HDMA reduction intermediates and products (red color).

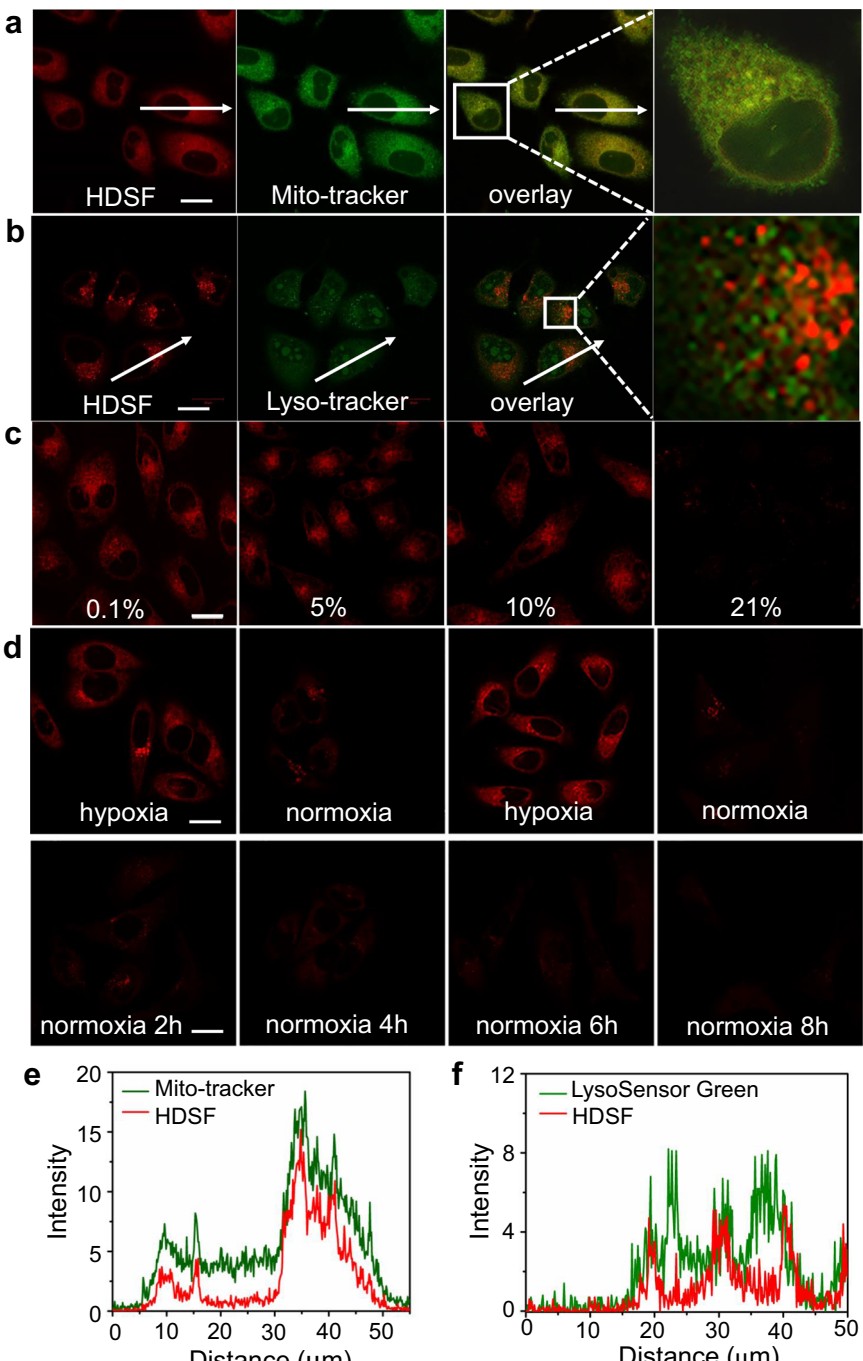

**Fig. 4 Mitochondria targeting, hypoxia imaging sensitivity, and reversibility in cells.** Confocal fluorescence images of MCF-7 cells co-stained with $2\,\mu M$ HDSF and Mito-Tracker Green (**a**) or HDSF and LysoSensor Green DND-189 (**b**) in hypoxia conditions (0.1% $O_2$). The red channel image was obtained with a band path of 640–750 nm upon excitation at 633 nm, and the green channel image was obtained with a band path of 492–630 nm upon excitation at 488 nm. Fluorescence profiles along the white arrow from the red channel and green channel are in diagram **e** (according to **a**) and diagram **f** (according to **b**). **c** Fluorescence images of HDSF-stained MCF-7 cells incubated in conditions with different $O_2$ contents. **d** Fluorescence images of MCF-7 cells stained with HDSF in hypoxia-normoxia cycles and in normoxia environment. The results are representative of three biologically independent experiments. Scale bars: $20\,\mu m$. Source data are available as a Source data file.

together with the probe's NIR emission encouraged us to evaluate the in vivo imaging ability for cycling hypoxia in living animal models such as zebrafish embryos and mice.

Zebrafish embryo is an ideal model for hypoxia research[53], successive BDM (2,3-butanedione monoxime) and water rinse treatment can offer fine cycling hypoxia model[54]. Microinjection of HDSF into the brain of zebrafish embryo (6-day-old) resulted in accumulation of HDSF in a certain round-shaped organ 18 h

post-injection. Confocal fluorescence imaging revealed that almost no fluorescence was observed in HDSF-injected zebrafish. BDM treatment of the embryo led to bright fluorescence signals within 5 min, and the fluorescence disappeared completely after rinsing with fresh water to remove BDM. The second BDM incubation triggered the strong fluorescence again, and subsequent removal of BDM quenched the fluorescence completely (Fig. 5). All these results confirmed that HDSF was able to

monitor drug-induced cycling hypoxia in the transparent zebrafish embryo via confocal fluorescence imaging. With transgenic zebrafish where certain organs were fluorescently marked, co-localization tests were carried out to figure out the round-shaped organ HDSF accumulated in (Supplementary Fig. 20). However, the fluorescence of HDSF didn't overlap with liver, pronephros, insulin, or exocrine pancreas marked with GFP or RFP, respectively. Based on the location of the organ and the fluorescent co-localization tests, we deduced the organ marked by HDSF probably to be the gallbladder.

With the NIR emission of this probe, in vivo optical imaging for tumor hypoxia was explored in mice xenografted with human breast MCF-7 tumor. HDSF was injected into normal tissue (Region of interest A, ROI A) and tumor tissue (ROI B, Fig. 6a). The fluorescence signal was observed in ROI B and increased

remarkably in 35 min post-injection (Fig. 6c). The temporal profile of fluorescence in ROI B revealed the process of probe reduction inside the tumor. However, no obvious fluorescent signal was detected in ROI A. This indicated that HDSF was able to sense tumor hypoxia effectively, and the distinct fluorescence in tumors provides HDSF the ability to discriminate tumor from normal tissue. In addition, intratumoral injection of HDSF demonstrated that the fluorescence intensity (20 min post-injection) in the larger tumor (~380 mm$^3$) was much brighter than that found in the smaller one (~150 mm$^3$, Fig. 6b, d). This data verified that HDSF was capable of distinguishing different sizes of solid tumors.

Cycling hypoxia has also been visualized in a simulated ischemia-reperfusion model by HDSF injected intramuscularly on the hind limbs of live mice (Fig. 7). Both limbs of the mouse exhibited weak fluorescence 20 min post HDSF injection. Then the right limb was treated with a tourniquet for 25 min to create an ischemia condition, resulting in a distinct fluorescence increment in the right limb (Fig. 7a–f). The temporal profiles of fluorescence suggested that the oxygen concentration decreased gradually due to the lack of blood supply (Fig. 7n). The tourniquet was removed at 25 min for reperfusion, and fluorescence dropped gradually in the following 35 min (Fig. 7g–m). In contrast, the fluorescence in the left limb (as a control) without tourniquet treatment showed almost no change due to the normal blood supply. The temporal profiles of fluorescence in the right limb demonstrated clearly the turn-on fluorescent response in ischemia condition and the subsequent turn-off behavior in the reperfusion process, confirming its real-time visualizing ability of HDSF for hypoxia in living mice. Since O$_2$ level resolved in blood altered during the ischemia-reperfusion process, the blood flow of the right limb was detected by Doppler Ultrasound. A strong blood flow signal was observed without any

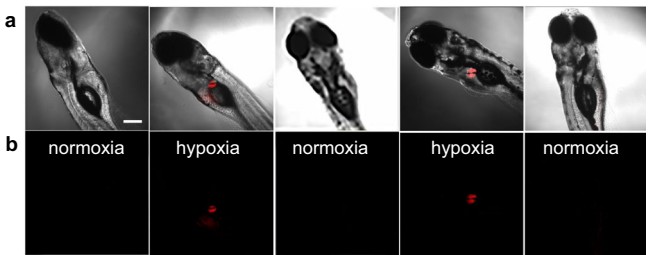

**Fig. 5 Reversible hypoxia imaging in zebrafish.** Confocal imaging of a 6-day-old zebrafish embryos injected with 2 nL HDSF (2 μM). Images were collected for zebrafish embryos underwent water rinse (normoxia) and BDM (15 mM, 5 min) treatment (hypoxia) successively in cycle. **a** Overlaps of bright field and fluorescence images, **b** fluorescence images. The results are representative of three biologically independent experiments. λ$_{ex}$, 633 nm. Scale bars: 200 μm.

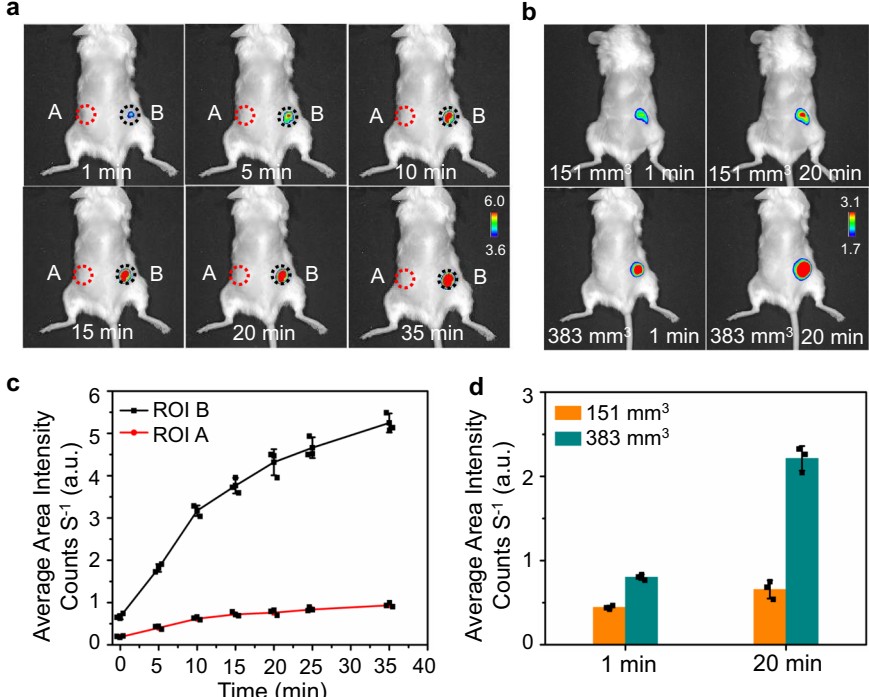

**Fig. 6 Tumoral hypoxia imaging in mice.** Optical imaging of human breast MCF-7 tumor-xenografted mice injected with HDSF (20 μM, 50 μL). **a** Images (overlaps of fluorescence and bright-field images) of a mouse underwent subcutaneously (ROI A) and intratumoral (ROI B) HDSF-injection recorded at different time post injection; **b** images of mice bearing tumors of different size (~151 and 383 mm$^3$) recorded at first or 20$^{th}$ min post intratumoral HDSF-injection; **c** temporal profile of fluorescence in ROIs A and B in (**a**); **d** histogram of fluorescence in tumors in (**b**). Data were presented as mean ± SD. The results are representative of three independent experiments. λ$_{ex}$, 660 nm; λ$_{em}$, 710 nm. Source data are available as a Source data file.

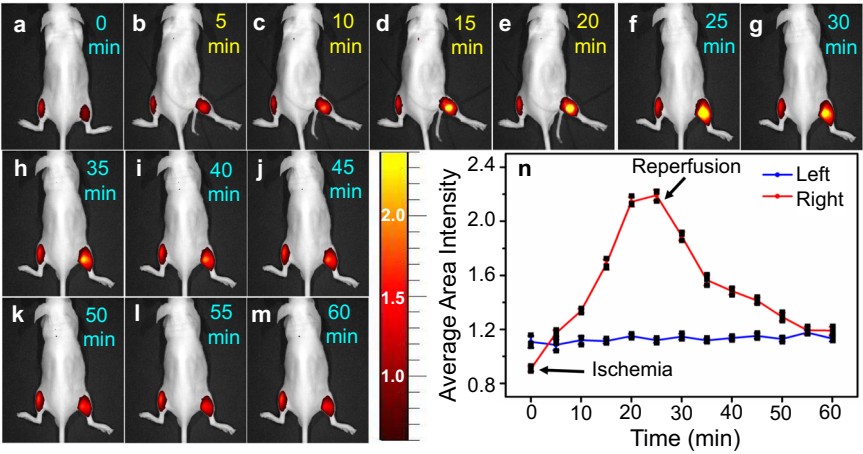

**Fig. 7 Hypoxia-reoxygenation imaging in mice hind limb.** Optical imaging of cycling hypoxia in ischemia-reperfusion process in a living mouse via intramuscular injection of HDSF (20 µM, 50 µL, 20 min) in the two hind limbs. The ischemia-reperfusion process was simulated via treating the right limb with a tourniquet for 25 min followed by removing the tourniquet thereafter. **a–m** images for mouse recorded every 5 min in the ischemia-reperfusion process (1 h); **n** temporal profiles for average fluorescence intensity in the left (blue) and right (red) limbs. Data were presented as mean ± SD. The results are representative of three independent experiments. $\lambda_{ex}$, 660 nm; $\lambda_{em}$, 710 nm. Source data are available as a Source data file.

treatment, which disappeared right after the limb was treated with a tourniquet. When the tourniquet was removed, the signal recurred (Supplementary Fig. 21, Supplementary Movie 1 collected 5 min before tourniquet treatment, Supplementary Movie 2 collected during tourniquet treatment, and Supplementary Movie 3 collected 5 min after tourniquet removal).

## Discussion

In conclusion, an arylazo-conjugated fluorescent NIR probe for cycling hypoxia in vivo, HDSF, was rationally constructed via combining 3,5-ditrifluoromethylbenzene with a xanthene/cyanine fused fluorophore via an azo-linker. Its reversible fluorescent response to hypoxia was not interfered by physiological pH and normal biorelated chemical species. Confocal imaging via HDSF-staining enabled sensitive imaging for cycling hypoxia in living cells and zebrafish embryos. More importantly, optical imaging of mice demonstrated that this probe was able to differentiate solid tumor size and track hypoxia-normoxia switch in ischemia-reperfusion process in living mice. All these attractive features profited from the modification of electron-withdrawing tri-fluoromethyl groups, which stabilized the phenylhydrazine intermediate and prevented the unwanted cleavage of the N–N bond. This study not only realized the real-time in vivo monitoring of cycling hypoxia in ischemia-reperfusion in mice, but also provided an effective design strategy for cycling hypoxia fluorescent probes.

Benefiting from this strategy, the development of hypoxia-responsive smart agents for precise tumor theranostics is undergoing in our group. For better quantification of the hypoxia level in biological samples, ratiometric probes with a reference fluorophore would be our future research direction. In addition, future efforts will also be devoted to the development of the second generation reversible hypoxia probes with improved repeat cycles, larger hypoxia/normoxia enhancement factors, and longer emission wavelength in the NIR II region.

## Methods

**Construction and characterization of HDSF and HDMA**. Details of synthesis and characterization of HDSF and HDMA can be found in SI Appendix (Supplementary Fig. 1–7).

**Spectroscopic study**. The stock solution of the probe was prepared in DMSO of HPLC pure grade and stored at −20 °C. Work solution of the probe was obtained

by diluting stock solution with PBS (0.1 M, pH 7.4). In vitro hypoxia tests were conducted in a glove box with $O_2$ content under 0.1 ppm, and normoxia tests were conducted in air. From normoxia to hypoxia, test solution was bubbled with pure argon gas until $O_2$ content was below 0.1 mg L$^{-1}$, and transferred to glove box for continued process. From hypoxia to normoxia, test solution was exposed into air until $O_2$ content was above 8 mg L$^{-1}$. For every hypoxia-normoxia cycle, RLM and NADPH were added repeatedly. Emission spectra were collected after 15 min when saturated by ambience, and absorption spectra were collected successively.

**Cell culture**. MCF-7 cells were purchased from National Collection of Authenticated Cell Cultures. Cells were cultured in MEM media with 10% FBS, 100 U mL$^{-1}$ penicillin, 100 µg mL$^{-1}$ streptomycin, and maintained in an incubator with a 5% $CO_2$ and 95% air atmosphere at 37 °C.

**Cytotoxicity assay**. Cytotoxicity of HDSF was tested with MCF-7 cells by MTT assay. Briefly, cells (4500 cells per well) were seeded in 96-well plates. Then cells were treated with 200 µL culture media containing different final concentrations of HDSF, and incubated for 12 h. MTT solution (20 µL, 5 mg mL$^{-1}$) was added to each well and incubated for 4 h at 37 °C. The supernatant was removed and DMSO (200 µL) was added to dissolve the MTT formazan. The plates were shaken in dark (10 min) until the formazan dissolved completely. Absorbance at 490 nm of each well was measured with a microplate reader (Thermo Scientific Varioskan Flash), and cell viability was calculated.

**Confocal fluorescence imaging of MCF-7 cells**. Images were acquired with confocal microscope Zeiss LSM710. The red imaging channel of HDSF was collected with a band path of 640–750 nm upon excitation at 633 nm, the green imaging channel of Mito-Tracker Green, LysoSensor Green DND-189, and Image-iT$^{TM}$ Green hypoxia reagent were collected with a band path of 492–630 nm upon excitation at 488 nm.

The hypoxic conditions (0.1, 5, and 10% $O_2$) were all generated by AneroPack® (Mitsubishi Gas Chemical Company, Inc.) with a matching culture bag (Mitsubishi Gas Chemical Company, Inc.). Cells sealed together with AneroPack® in a culture bag, was incubated in an incubator with a 5% $CO_2$ and 95% air atmosphere at 37 °C. In hypoxia-normoxia cycle, MCF-7 cells were first incubated with 2 µM HDSF at 37 °C for 2 h under hypoxia (0.1% $O_2$) and imaged, then incubated in a 5% $CO_2$ and 95% air atmosphere at 37 °C and imaged.

For the HDSF and commercial hypoxia reagent co-staining experiment, MCF-7 cells were incubated with 2 µM HDSF and 5 µM Image-iT$^{TM}$ Green hypoxia reagent at 37 °C for 2 h under hypoxia (0.1% $O_2$) and imaged, then incubated in a 5% $CO_2$ and 95% air atmosphere at 37 °C and imaged.

For the intracellular distribution experiment, MCF-7 cells were incubated with 2 µM HDSF under hypoxia (0.1% $O_2$) at 37 °C for 2 h. Then 50 nM Mito-Tracker Green or 1 µM LysoSensor Green DND-189 was added and incubated under hypoxia (0.1% $O_2$) for another 1 h before imaging. The dual channel imaging mode was adopted. The Pearson's correlation coefficients were reported by Zen software on confocal microscope Zeiss LSM710.

**Confocal fluorescence imaging of zebrafish stained by HDSF**. The 6-day-old zebrafish embryos used in cycling hypoxia imaging tests were wild type Tubingen strain (TU), gifts from Prof. Zhao Qingshun Group of Model Animal Research

Center (NJU). Transgenic zebrafish embryos used in co-localization tests were obtained from China Zebrafish Resource Center.

Images were acquired with confocal microscope Zeiss LSM710 system. The red imaging channel of HDSF was collected with a band path of 640–750 nm upon excitation at 633 nm, the green imaging channel of GFP was collected with a band path of 500–600 nm upon excitation at 488 nm, and the bule imaging channel of RFP was obtained with a band path of 550–700 nm upon excitation at 543 nm.

In hypoxia-normoxia cycle tests, zebrafish was microinjected with HDSF (2 μM, 2 nL) in the brain, and maintained in E3 medium at 28 °C for 18 h before imaging. BDM (2,3-butanedione monoxime, 15 mM, 1% DMSO, v/v) was applied to incubate zebrafish for 5 min to abolish its cardiac contraction and create a hypoxic condition. Tricaine (0.042 mg mL$^{-1}$) was used as anesthesia for embryo incubation (~30 s) and images under hypoxia were taken. By rinsing with fresh-water, the cardiac contraction was recovered and oxygen level was reversed to normal in about 2 h, and images under normoxia were recorded. These processes were repeated again for the second hypoxia-normoxia cycle.

In fluorescent co-localization tests, 6-day-old transgenic zebrafish embryos were microinjected with HDSF (2 μM, 2 nL) in the brain, and maintained in E3 medium at 28 °C for 18 h before imaging. The hypoxia condition was created with BDM incubation.

**Acute toxicity of HDSF.** Eight-week-old female ICR mice were obtained from Model Animal Research Center (NJU), housed in polycarbonate cages at 22–25 °C, 35–45% humidity with standard food and water supply and 12 h light-darkness cycles. Mice were divided into five groups randomly, injected 50 μL saline (control) or saline containing various concentrations of HDSF (2% DMSO as cosolvent), and body weights were recorded for 7 days. Then mice were sacrificed, main organs (heart, liver, spleen, lung, and kidney) were collected for hematoxylin and eosin (H&E) staining. Images were collected with a Nikon DS-Fi3 microscope.

**Optical imaging of mice with HDSF injection.** For hypoxia imaging in solid tumors, the NCG MCF-7 tumor-bearing mice supplied by Model Animal Research Center (NJU) were adopted. The shaved mice were then injected with HDSF (20 μM, 50 μL) in normal tissue (the lower back, subcutaneous injection) and tumor (intra-tumorous injection), respectively. Isoflurane was used as anesthesia. Imaging was performed on a PerkinElmer IVIS Lumina K Series III in vivo imaging system, and images of mice in prone position were collected at 710 nm upon excitation at 660 nm.

For ischemia-reperfusion visualization, male adult nude mice from Model Animal Research Center (NJU) were used. HDSF (50 μL, 12.5 μM) was intramuscularly injected into the two hind limbs. Optical image was recorded 20 min post HDSF injection (Fig. 7a). Then the right hind limb was tightly bound with a tourniquet for 25 min to create the ischemia condition in the limb (Fig. 7b–f). Then the tourniquet was removed for subsequent imaging (Fig. 7f–m). The imaging was carried out after HDSF injection and images of mice in prone position were recorded every 5 min.

For Doppler Ultrasound imaging in the ischemia-reperfusion process, mice were injected with 5% chloral hydrate in the abdomen. Blood flow of the limb was limited with a tourniquet, then 5 min later the tourniquet was moved away to create an ischemia-reperfusion process. Blood flow videos of the right hind limb were collected before (5 min), during and after (5 min) the process. Videos are supplied in SI.

**Ethics statement.** All animal tests were conducted according to the Guidelines for the Care and Use of Laboratory Animals of the Chinese Animal Welfare Committee and approved by the Institutional Animal Care and Use Committee of Nanjing University.

**Data analysis.** Average area intensities in cells were given by the Zen software on confocal microscope Zeiss LSM710. Average area intensities in mice were given by the Living Image software (version 4.5) on PerkinElmer IVIS Lumina K Series III in vivo imaging system. All data were analyzed and calculated with Microsoft Excel 2016 software (Microsoft, Redmond, WA), and the statistical differences were analyzed by a two-tailed student's test. All statistical data were presented as means ± SD. All statistical graphs and fluorescent spectra were performed using Origin 8.5 (OriginLab Corporation, MA, USA). Quantum chemistry calculation was performed with Gaussian 0.9, revision D.01 (Gaussian Inc., Wallingford, CT, 2013).

**Reporting summary.** Further information on research design is available in the Nature Research Reporting Summary linked to this article.

## Data availability

The authors declare that all data supporting conclusions of this work are available either in the paper or in the supporting information files or from the authors upon request. Source data are provided with this paper.

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

## Acknowledgements

The work was under financial supports from the National Natural Science Foundation of China (Grant Nos: 21977044, 21907050, 21731004, and 91953201), the Natural Science Foundation of Jiangsu Province (BK20190282 and BK20202004), and the Excellent Research Program of Nanjing University (ZYJH004). We thank Prof. Deju Ye, Prof. Xin Lou from Nanjing University and Prof. Jinrong Peng from Zhejiang University for constructive discussions.

## Author contributions

Y.Z., Y.C., Z.G., and W.H. designed the study. Y.Z. and S.Y. synthesized probes; W.Z., Z. L., and Y.L. performed the DFT calculations. Y.Z., H.Y., and H.F. performed the cellular and zebrafish experiments; Y.Z., C.Z., and H.X. performed the in vivo experiments; N.L. and Q.Z. provided the zebrafish and co-designed the zebrafish tests. Y.Z. and Y.C. co-wrote the manuscript. All authors discussed the results and commented on the paper. All authors have given approval to the final version of the manuscript.

## Competing interests

The authors declare no competing interests.
