## [Peer Review File · Nature Communications]

REVIEWER COMMENTS

Reviewer #1 (Remarks to the Author): Expert in probe synthesis

This manuscript reported a reversible arylazo-based NIR probe for cycling hypoxia imaging in vivo. This work is very interesting in the field of hypoxic fluorescent probe. The manuscript can be accepted for publication after the following points are addressed.

1. In Fig 1, Fig S7,8, it is better to give the background UV-Vis and fluorescence spectra of the system (PBS+ RLM+NADPH without HDMA or HDSF)
2. HDSF is a new compound used for the fluorescence probe, the arylazo part can efficiently quench the fluorescence of xanthene/cyanine fused fluorophore. How about the exact quenching mechanism? FRET, PET or some other process? The authors didn't give the whole UV-Vis spectra of HDSF. How about the absorption band of the azobenzene part?
3. In Fig S7, from normoxia to hypoxia, N=N bond will change to NH-NH, how to explain the increasement of absorption peak of HDMA and the red shift of HDSF? The reduction of azo bond can be confirmed by the UV-Vis spectra change. The authors should give the whole UV-Vis spectra change from 300 to 850 nm.
4. In Figs7, for emission spectra, RLM is 250 ug/mL, NADPH is 100 uM, while for absorption spectra, RLM is 25 ug/mL, NADPH is 40 uM. Why? The same example should be used for emission and absorption measurement.
5. In Fig2, Figs14, the ESI-MS spectrum is measured by the origin reaction solution or treated by some process?
6. From Figs8, it seems that the reduction reaction has saturated after 15 min. Anyway, In Fig2a, after 15 min reaction, a large amounts of HDSF still exist in the system. How about the extent of reduction reaction?
7. Some recent papers about the azobenzene based fluorescent probe for hypoxia imaging can be cited, such as Chem. Eur. J. 2019, 25, 9634; Chem. Eur. J., 2020, 26, 2521; Nanoscale, 2020, 12, 7509.

Reviewer #2 (Remarks to the Author): Expert in zebrafish and hypoxia

The manuscript of Zhang et al. describes the generation of a reversible oxygen probe. This may become a powerful tool in in vivo analysis of cycling hypoxia, although some chemicals have been produced this is a novel class that might be less toxic. In principle this appears interesting and exciting but there are a number of issues with the manuscript.

One of the advantages is according to the authors a reduced toxicity, but studies showing levels of how much an animal can tolerate are not given. If this feature clearly sets this probe apart, and above others, they should provide data for this.

The paper is written stressing the positives, if this is to be adopted by others I always welcome authors telling me honestly and clearly what the issues are with their "pet molecule". What are the as yet unsolved problems, according to the authors?

In my opinion, one of these is the absence of a control for accumulation, although I will be the first to admit that on/off probes are much more convenient than ratiometric ones, this is not even

brought up in the paper and this could be a real issue. For instance, tumours could accumulate the compound.

There may be other problems with it, for instance 3 cycles of hypoxia/reoxygenation are shown, how far can one go, is 10 cycles possible? How stable is the critical bond in the compound under continuous normoxia and hypoxia? Some detailed measurements are needed to show those limits.

The accumulation problem may actually become clear in Figure 4. The authors state there is accumulation of fluorescence in the heart, however, the image is so small this cannot be properly seen. Importantly, the bright red spot sits in a position that I would judge to be the glomeruli/pronephros rather than the heart. Could it be that the fluorescent molecule (or its precursor) accumulates in the excretory system? We have observed similar things with fluorescein/rhodamine.

An exciting alternative would be that the embryonic kidney is the first organ where oxygen runs out if the heart is stopped by BDM. Oxygen diffusion is rapid in fish as they are tiny, the kidney is located deep in the centre of the embryo, and as in humans, it might be a site where low physiological oxygen levels are "noticed" very early.

In these experiments the molecule was injected in the brain and is supposed to diffuse to other areas, but there is no proof of this distribution.

A useful control to show the overall distribution, would be to put the larvae in a strongly hypoxic environment, this should light up the probe everywhere, and could show even/uneven distribution of the chemical.

In mice the experiments involve injection into the tumor. There is more fluorescence in the larger tumour, but does it simply retain more of the compound?

Ideally the authors should try to create a situation where compound is distributed evenly and show that the level of signal at least roughly correlates with tissue hypoxia as shown with a different marker (eg hypoxyprobe).

Small remarks

Overall the manuscript would benefit from some editing for english language

Figure 3 needs a blow-up to make overlap between mitotracker and HDSF clearer

Fig S16 is not convincing, this may be because HIF is extremely unstable, it may be better to do qPCR for a good HIF reporter eg EGLN3/PHD3 or LDHA

Typos

The fluorescence increased significantly with the decrement of O₂ content in incubation environment. ...decrement, environment

various reductases, and NADPH as electron donor.[14b, 15a]..... donor

cycle, MCF-7 cells was first incubated with 2?M HDSF at 37oC. ... odd question mark

Freek van Eeden

Reviewer #3 (Remarks to the Author): Expert in probe synthesis

The authors in this paper constructed a reversible arylazo-conjugated fluorescent NIR probe, HDSF, for cycling hypoxia *in vivo*, by combining 3,5-ditrifluoromethylbenzene with a xanthene/cyanine fused fluorophore via an azo-linker. Confocal imaging via HDSF-staining enabled sensitive imaging for cycling hypoxia in living cells and zebrafish embryos. Furthermore, this probe was able to differentiate hypoxia level in solid tumors and track hypoxia-normoxia switch in ischemia-reperfusion process in living mice. All these attractive features were profited from the modification of electron-withdrawing trifluoromethyl groups, which stabilized the phenylhydrazine intermediate and prevented the unwanted cleavage of N-N bond. This provided an effective design strategy for cycling hypoxia fluorescent probes. I think this work is well done and can be published on Nat. Commun. after solving the following problems.

- 1) In the selectivity studies, how about the interference of $\bullet\text{OH}$ and ONO_2 -?
- 2) For the *in vivo* imaging of tumor hypoxia, how about the longer duration of the probe in tumor site.
- 3) How about the cytotoxicity of the probe?
- 4) All the synthesized compounds should be characterized, at least NMR should be provided.
- 5) There are some spelling mistakes need to be corrected, for example, "monitered", "donnor" in page 2; "invesigate" in page 3; "drement" and "environmnet" in page 4.
- 6) "*in vivo*" should be italic.

Reviewer #4 (Remarks to the Author): Expert in hindlimb ischemia

In the manuscript, the authors constructed a reversible arylazo-conjugated fluorescent probe HDSF to detect cycling hypoxia. They explain the chemical mechanism of the reversibility by oxygen levels. They show the *in vivo* use of the reagent by drug-induced hypoxia in the zebrafish embryo, in tumor xenograft in mice, and ischemia-reperfusion of the mouse limb. It is intriguing since the probe is reversible and can detect hypoxia in the tumor in mice. The authors claim HDSF is the first organic fluorescent probe capable of reversibly real time imaging of drug-induced and mechanical force induced hypoxia *in vivo*.

However, there are some questions and concerns.

There are many data in the supplement (up to 19), including important information. Some data can be integrated into the main text. For example, Figure S9 with Figure 1, and Figure S14 with Figure 2 can be combined to make a clear contrast.

In Figure 6, it looks like the probe detects mild intramuscular hypoxia before the treatment in both legs. The tourniquet-induced ischemia in 15 min increases the hypoxic signal, suggesting the response to the gradient of O_2 level. But Figure 3 and Figure S17 show that the probe detects mild hypoxia as 10% O_2 in cells and does not show much difference between 10% and 0% O_2 . How is it explained?

Figure 6 indicates the local injection of the probe does not expand and stays locally. Does systemic injection (intra-vascular) detect the whole ischemic limb during shutting blood flow? How long may it last *in vivo*?

Why were the nude mice used for ischemia-reperfusion limb study? Does the probe cause immunoreactivity?

There are 3 videos in the supplement, but no explanation in detail, except it looks like Doppler ultrasound images of the ischemia-reperfusion process of mouse limb.

Figure S13. Are DMPO only (a) and control without DMPO (b) in hypoxia? Does it mean hypoxia generates radicals, but hypoxic HDSF may not? In the text, “This data demonstrated that the single electron reduction product ...of HDSF was not stable enough..., suggesting a new mechanism different from previously reported arylazo-reduction based hypoxia probes”. But later with Fig S17, they state, “fluorescence increment under hypoxia was a result of HDSF reduction.” It is confusing. Figure S17. Dicoumarol is a Vitamin K-relating reductase inhibitor. DPI is a flavoenzyme inhibitor that may quench superoxide radicals. Please adjust the interpretation. Does this suggest HDSF reacts to superoxide radicals?

A reversible two-photon luminescent probe has been reported to show cycling hypoxia in cells and zebrafish (Zhang P, 2015, PMID25890748), and various hypoxia probes are clinically used to trace tumors (Mirabello, review 2018 PMID29527524). The novelty does not appear so strong as the authors claim. They should explain the uniqueness and advantage of using their probe.

Minor points:

Figure S16. “HIF-1a level in cell supernatant” must be in “cell lysate.” HIF-1a is not secreted protein. Western blot does not indicate molecular size. It is hard to appreciate the HIF-1 induction by hypoxia compared to normoxia.

They should explain the acronym when it appears first in the text for general audiences. (e.g., NIR, HDSF, DFT, EST-MS, ESI). Also, Scheme 1 is hard to understand without much explanation.

The last part of the Methods should be about Figure 6, not Figure 5.

The references are in an odd format. It should list only one publication with each number.

RESPONSE TO REVIEWER COMMENTS

Reviewer #1:

Q: This manuscript reported a reversible arylazo-based NIR probe for cycling hypoxia imaging in vivo. This work is very interesting in the field of hypoxic fluorescent probe. The manuscript can be accepted for publication after the following points are addressed.

A: Thanks for the valuable comments and suggestions, we have prepared a detailed point to point response below.

Q1. In Fig 1, Fig S7,8,9 it is better to give the background UV-Vis and fluorescence spectra of the system (PBS+ RLM+NADPH without HDMA or HDSF)

A: Thanks for this suggestion, the absorption and emission of PBS buffer containing RLM and NADPH (probes absent) have been added to the spectra respectively.

Q2. HDSF is a new compound used for the fluorescence probe, the arylazo part can efficiently quench the fluorescence of xanthene/cyanine fused fluorophore. How about the exact quenching mechanism? FRET, PET or some other process? The authors didn't give the whole UV-Vis spectra of HDSF. How about the absorption band of the azobenzene part?

A : The quenching mechanism is that arylazos exhibit large non-radiative decay rates (\$k_{nr}\$ ) due to the ultrafast photo-induced E-Z isomerization (references of [40, 44, 45] in the manuscript), which makes these arylazo-conjugated probes almost non-fluorescent. The whole absorption spectra of HDMA and HDSF from 300-800 nm have updated and shown in supplementary Fig. 7 (Fig. R1). Electron donating group modification and longer conjugation length will cause red-shifting of the absorption spectra of azobenzene derivatives (reference: Rau, H. Photoisomerization of azobenzenes. *Photoreactive Organic Thin Films* **34**, 2003, 3-47). Since the azobenzene part was conjugated with the merocyanine moiety, the newly formed large conjugated \$\pi\$ system showed a broad absorption band at longer wavelength with shoulder bands. It was hard to distinguish the characteristic absorption of the azobenzene part from that of the whole molecule.

Fig. R1 Absorption spectra of HDMA and HDSF in PBS buffer, (a) without the existence of RLM and NADPH, (b, c) with the existence of RLM and NADPH. Ctrl: PBS with RLM and NADPH.

Q3. In Fig S7, from normoxia to hypoxia, N=N bond will change to NH-NH, how to explain

the increasement of absorption peak of HDMA and the red shift of HDSF? The reduction of azo bond can be confirmed by the UV-Vis spectra change. The authors should give the whole UV-Vis spectra change from 300 to 850 nm.

A: As shown in Fig R1, in hypoxia condition, the N=N bond was reduced to NH₂ for HDMA and NH-NH for HDSF, respectively. Since the electron donating abilities of NH₂ and NH-NH groups are stronger than that of N=N group, the intramolecular charge transfer (ICT) effects of the reduced products HD-NH₂ and HDSF-HZ should be stronger than those of HDMA and HDSF. Therefore, the absorption band centered at 693 nm is identical to band of HD-NH₂, which is red-shifted compared to that of HDMA. Similar result was observed for HDSF, a red shifted absorption band centered at about 685 nm is due to the enhanced ICT effect after reduction.

Q4. In Figs7, for emission spectra, RLM is 250 ug/mL, NADPH is 100 uM, while for absorption spectra, RLM is 25 ug/mL, NADPH is 40 uM. Why? The same example should be used for emission and absorption measurement.

A: This is a good comment, and the concentration of RLM and NADPH have been adjusted to the same in absorption and emission tests.

Q5. In Fig2 , Figs14, the ESI-MS spectrum is measured by the origin reaction solution or treated by some process?

A: To avoid rat liver microsomes blocking EMI-MS machine, spectra were collected using supernatant of the origin reaction solution (Fig. 2a) and extract solution of reaction solution (Fig. 2b and 2c).

Q6. From Figs8, it seems that the reduction reaction has saturated after 15 min. Anyway, In Fig2a, after 15 min reaction , a large amounts of HDSF still exist in the system. How about the extent of reduction reaction?

A: Indeed, when fluorescent intensity saturated after 15 min, the reaction reached a balance with most of HDSF reduced and others unchanged. However, since HDSF is a reversible probe, the reduced product HDSF-HZ could be easily re-oxidized to HDSF under normoxia condition. During the ESI-MS test, it was hard to keep the solution in a hypoxia environment, thus some HDSF-HZ may return to HDSF. These factors all lead to obvious MS of HDSF in the ESI-MS spectra of Fig 2a.

Moreover, HPLC test was conducted to evaluate the extent of reduction reaction. During HPLC test, although it was also difficult to cut air off completely, the results showed that most of HDSF was converted to HDSF-HZ under hypoxia for 15 min with the area ratio of HDSF-HZ/HDSF nearly 5/1 (supplementary Fig. 14, Fig. R2).

Fig. R2 HPLC spectra of HDSF solution (20 μM in PBS buffer, pH 7.4) incubated with RLM (250 $\mu\text{g}/\text{mL}$) and NADPH (100 μM) at 37°C for 15 min (a) in air, (b) in hypoxic condition of glove box, (c) in air after incubated in hypoxia condition. Elution solvent: mixture of methanol and water (v/v = 1/1) containing 0.1% TFA. Data collected at 680 nm.

Q7. Some recent papers about the azobenzene based fluorescent probe for hypoxia imaging can be cited, such as Chem. Eur. J. 2019, 25, 9634; Chem. Eur. J., 2020, 26, 2521; Nanoscale, 2020, 12, 7509.

A: Thanks for the suggestion, some new and interesting hypoxia probes reported recently have been added to references.

Reviewer #2 (Remarks to the Author): Expert in zebrafish and hypoxia

Q: The manuscript of Zhang et al. describes the generation of a reversible oxygen probe. This may become a powerful tool in in vivo analysis of cycling hypoxia, although some chemicals have been produced this is a novel class that might be less toxic. In principle this appears interesting and exciting but there are a number of issues with the manuscript.

A: Thanks for the comments, we have prepared a point by point response below.

Q1. One of the advantages is according to the authors a reduced toxicity, but studies showing levels of how much an animal can tolerate are not given. If this feature clearly sets this probe apart, and above others, they should provide data for this.

A: Cytotoxicity of HDSF in MCF-7 cells was added (supplementary Fig. 15, Fig. R3), the results showed that the viability of MCF-7 cells was over 80% with incubation concentration up to 4 \$\mu\text{M}\$. All the cellular imaging studies were conducted at 2 \$\mu\text{M}\$, which will not show significant toxicity. Furthermore, toxicity in mice was tested. 25 female ICR mice were divided randomly into five groups, and 50 \$\mu\text{L}\$ saline solution containing different concentrations of HDSF (0-400 \$\text{mg kg}^{-1}\$ ) was injected respectively. Body weight was recorded in the next 7 days (supplementary Fig. 21, Fig. R4), and then main organs (heart, liver, spleen, lung, kidney) were collected for H&E staining (supplementary Fig. 22, Fig. R5). No obvious body weight lost or organ damage were found.

Fig. R3 MCF-7 cell viability after treated with different concentrations of HDSF for 12 h, measured by MTT assay. Data are mean \pm s.d.(n=3).

Fig. R4 Body weight changes of mice over 7 days. Mice were injected with 50 μ L saline (control) or HDSF saline solution at the first day. Data are mean \pm s.d.(n=5).

Fig. R5 H&E staining of main organs of mice injected with saline (Ctrl) or HDSF solution (20 mg kg⁻¹, 60 mg kg⁻¹, 200 mg kg⁻¹, 400 mg kg⁻¹, respectively) at the first day and sacrificed after 7 days for H&E staining tests. Scale bar: 20 μ m.

Q2: The paper is written stressing the positives, if this is to be adopted by others I always welcome authors telling me honestly and clearly what the issues are with their “pet molecule”. What are the as yet unsolved problems, according to the authors?

A: Thanks for this comment. Actually, although we emphasized on the advantages of

HDSF, there were indeed some problems needed to be solved or improved in the future study. For example, the fluorescence enhancement factor was only 6-7 folds after HDSF was treated under hypoxia environment. However, our second generation hypoxia probe which is currently under initial investigation showed ~96 fold fluorescence enhancement in hypoxia. In addition, it is hard for the single wavelength intensity change based probe (HDSF) to quantify hypoxia, since other factors such as probe concentration would easily interfere the fluorescence intensity. In this regard, our future efforts will be devoted to develop ratiometric hypoxia probes overcome the problem. Moreover, to further improve the *in vivo* imaging depth, developing NIR-II based hypoxia probes will be another future direction. We have discussed this in the manuscript.

Q3: In my opinion, one of these is the absence of a control for accumulation, although I will be the first to admit that on/off probes are much more convenient than ratiometric ones, this is not even brought up in the paper and this could be a real issue. For instance, tumours could accumulate the compound. There may be other problems with it, for instance 3 cycles of hypoxia/reoxygenation are shown, how far can one go, is 10 cycles possible? How stable is the critical bond in the compound under continuous normoxia and hypoxia? Some detailed measurements are needed to show those limits.

A: Thanks for the comment. Indeed, tumors could accumulate the compound. For the control of accumulation, the ROI A in Fig.5 could be viewed as a control, while the left hind limb in Fig. 6 was set as a control. Since the compound is non-fluorescence in normoxia tissue, it is difficult to evaluate the accumulation of HDSF. To solve this problem, we plan to synthesize a ratiometric sensor by attaching a reference fluorophore.

As suggested by the reviewer, we did further measurement to testify stability of the critical hydrazine bond, nearly 10 hypoxia cycles were measured (supplementary Fig. 9, Fig. R6). During the first several normoxia-hypoxia cycles, fluorescence on-off phenomenon repeated well. However, from the fifth cycle, fluorescence couldn't absolutely return to the original level in normoxia condition and the base line shifted. It is probably due to breakdown of the critical hydrazine bond during the repeated normoxia-hypoxia cycles. Future efforts would also be devoted to improve hypoxia/reoxygenation repeat cycles.

Fig. R6 Emission intensity change at 705 nm when HDSF buffer solution containing RLM and NADPH incubated at 37°C in normoxia (in air)-hypoxia (in glove box, 15 min) cycles.

Q4: The accumulation problem may actually become clear in Figure 4. The authors state

there is accumulation of fluorescence in the heart, however, the image is so small this cannot be properly seen. Importantly, the bright red spot sits in a position that I would judge to be the glomeruli/pronephros rather than the heart.

Could it be that the fluorescent molecule (or its precursor) accumulates in the excretory system? We have observed similar things with fluorescein/rhodamine.

An exciting alternative would be that the embryonic kidney is the first organ where oxygen runs out if the heart is stopped by BDM. Oxygen diffusion is rapid in fish as they are tiny, the kidney is located deep in the centre of the embryo, and as in humans, it might be a site where low physiological oxygen levels are “noticed” very early.

A: Thanks for the nice suggestion. Transgenic zebrafish embryos were purchased from China Zebrafish Resource Center. In the embryos, special organs including liver, pronephros, insulin, exocrine pancreas are marked with GFP or RFP respectively (supplementary Fig. 19, Fig. R7). Unfortunately, gallbladder labeled embryos were not available from the China Zebrafish Resource Center. The results of co-localization study showed that none of fluorescence of these marked organs overlapped with that of HDSF. However, based on our data and the anatomy image of a 5-dpf zebrafish embryo (reported by Wallace, K.N., and Pack, M., supplementary Fig. 19g), we deduced the organ marked by HDSF to be gallbladder.

Fig. R7 Confocal imaging of transgenic zebrafish embryos injected with HDSF (2 μ M) and incubated with BDM (15 mM, 5 min) before imaging. Fluorescent protein (GFP or RFP) labeled organs in embryos respectively, liver (a-d), pronephros (e-h), insulin (i-l), exocrine pancreas and liver (m-p). (q) Top: zebrafish anatomy; Bottom: the enlarged image of (p).

Q5: In these experiments the molecule was injected in the brain and is supposed to diffuse to other areas, but there is no proof of this distribution.

A useful control to show the overall distribution, would be to put the larvae in a strongly hypoxic environment, this should light up the probe everywhere, and could show even/uneven distribution of the chemical.

[A: We tried to create a severe hypoxia environment with AneroPack[®] of 5-10% O₂ \(Mitsubishi Gas Chemical Company, Inc.\). Unfortunately, almost no larvae could survive in such an environment. Developing a ratiometric probe by fusion with a reference fluorophore will solve this problem, which is undergoing in our lab.](#)

Q6: In mice the experiments involve injection into the tumor. There is more fluorescence in the larger tumour, but does it simply retain more of the compound?

Ideally the authors should try to create a situation where compound is distributed evenly and show that the level of signal at least roughly correlates with tissue hypoxia as shown with a different marker (eg hypoxyprobe).

A: Thanks very much for this comment. Indeed, since HDSF is an on-off probe, the fluorescence intensity is correlate to several factors including probe concentration and O₂ content. It is possible that more compound accumulation and more intense hypoxia condition both contribute to the stronger fluorescence of larger tumor. However, since it took time for HDSF to spread, at the beginning of injection, almost equal concentration of HDSF existed in small and large tumours. Thus, within 1 min after HDSF injection, fluorescence intensity difference may be mainly originated from difference of hypoxia condition. Again, ratiometric hypoxia probes are expected to resolve this problem and to better differentiate hypoxia extent in tumors. We have changed our statement to a more "conservative" tone in the manuscript.

As to the distribution, it is really hard to guarantee for compound to distribute evenly in tumour because of the complicated inner tumoral environment. We also tried intravenous

injection, however no obvious fluorescence signal was found in tumour over a period of 24 h post injection.

Small remarks

Q7: Overall the manuscript would benefit from some editing for english language.

A: Thanks, we have ask some experienced scholars to help with the editing and English language.

Q8: Figure 3 needs a blow-up to make overlap between mitotracker and HDSF clearer.

A: Thanks for the suggestion, we have added the zoom-in picture for higher clarity.

Q9: Fig S16 is not convincing, this may be because HIF is extremely unstable, it may be better to do qPCR for a good HIF reporter eg EGLN3/PHD3 or LDHA.

[A: Thanks for your comment. HIF test was supposed to verify the effectivity of cellular hypoxia-normoxia cycle created by AneroPack[®] and air-exposure. We took an alternative way to confirm the hypoxia environment using a commercial hypoxia fluorescent probe to co-stain cells with HDSF in hypoxia-normoxia cycle to prove the effectivity. As shown in supplementary Fig. 16 \(Fig. R8\), no obvious fluorescence appeared in normoxia cells. After cells treated in AneroPack[®] \(Mitsubishi Gas Chemical Company, Inc.\), both green fluorescence \(Hypoxia reagent Image-iT[™] Green\) and red fluorescence \(HDSF\) appeared, revealing a hypoxia environment in cells and HDSF was able to image it. When cells exposed to air and O₂ content increased, red fluorescence of HDSF dimmed, and green fluorescence of Hypoxia reagent Image-iT[™] Green stayed still because of the irreversibility.](#)

Fig. R8 MCF-7 cells co-stained with HDSF and Image-iT[™] Green Hypoxia Reagent in hypoxia-normoxia cycles. Red channel was obtained with a bandpath of 640-750 nm upon excitation at 633 nm, and green channel was obtained with a bandpath of 492-630 nm upon excitation at 488 nm. Scale bars: 20 μ m.

Q10: Typos

The fluorescence increased significantly with the decrement of O₂ content in incubation environment. ...decrement, environment various reductases, and NADPH as electron donor.[14b, 15a]..... donor

cycle, MCF-7 cells was first incubated with 2?M HDSF at 37oC. ... odd question mark

A: Thanks very much, we have corrected the mistakes.

Reviewer #3

Q: The authors in this paper constructed a reversible arylazo-conjugated fluorescent NIR probe, HDSF, for cycling hypoxia in vivo, by combining 3,5-ditrifluoromethylbenzene with a xanthene/cyanine fused fluorophore via an azo-linker. Confocal imaging via HDSF-staining enabled sensitive imaging for cycling hypoxia in living cells and zebrafish embryos. Furthermore, this probe was able to differentiate hypoxia level in solid tumors and track hypoxia-normoxia switch in ischemia-reperfusion process in living mice. All these attractive features were profited from the modification of electron-withdrawing trifluoromethyl groups, which stabilized the phenylhydrazine intermediate and prevented the unwanted cleavage of N-N bond. This provided an effective design strategy for cycling hypoxia fluorescent probes. I think this work is well done and can be published on Nat. Commun. after solving the following problems.

A: Thanks for the valuable words, we have prepared a point by point response below.

Q1: In the selectivity studies, how about the interference of •OH and ONO₂⁻?

A: The interference of O₂⁻, •OH and ONO₂⁻ were tested and added in supplementary Fig. 11b (Fig. R9). No obvious influence on HDSF emission was observed.

Fig. R9 Histogram of HDSF fluorescence at 705 nm in the presence of 100 μM sodium nitroferricyanide (III) (SNP), NO₂⁻, OCl⁻, H₂O₂, HS⁻, Hcy, GSH, Cys, ascorbic acid (asc), oxalic acid (oxa), •OH, 20 μM O₂⁻, and ONO₂⁻ respectively.

Q2: For the in vivo imaging of tumor hypoxia, how about the longer duration of the probe in tumor site.

A: Aliquot HDSF solution (20 μM, 50 μL) was injected into tumours with different size (166 mm³, 380 mm³), and images of mice were recorded over 24 h post injection (supplementary Fig. 23, Fig. R10). Fluorescence intensity of HDSF increased and kept steady over 12 h post injection, with a little fluctuation around 6 h in the large tumour.

Fig. R10 Temporal profile of average fluorescence intensity in tumour with different size after intratumorous HDSF-injection (20 μ M, 50 μ L). The results are presented as the mean \pm SD, n = 3 biologically independent mice per group. λ_{ex} , 660 nm; λ_{em} , 710 nm.

Q3: How about the cytotoxicity of the probe?

A: Cytotoxicity of HDSF in MCF-7 cells was added (supplementary Fig. 15, Fig. R3), the results showed that the viability of MCF-7 cells was over 80% with incubation concentration up to 4 \$\mu\$ M. All the cellular imaging studies were conducted at 2 \$\mu\$ M, which will not show significant toxicity. Furthermore, toxicity in mice was tested. 25 female ICR mice were divided randomly into five groups, and 50 \$\mu\$ L saline solution containing different concentrations of HDSF (0-400 mg kg⁻¹) was injected respectively. Body weight was recorded in the next 7 days (supplementary Fig. 21, Fig. R4), and then main organs (heart, liver, spleen, lung, kidney) were collected for H&E staining (supplementary Fig. 22, Fig. R5). No obvious body weight lost or organ damage were found.

4) All the synthesized compounds should be characterized, at least NMR should be provided.

A: As the synthesis and characterization of HD-NH₂ was already reported by X. He and coworkers in 2007 (supplementary reference 2), thus only ¹H NMR data is added in supplementary information.

5) There are some spelling mistakes need to be corrected, for example, "monitered", "donnor" in page 2; "invesigate" in page 3; "drement" and "environmnet" in page 4.

A: Thanks, we have corrected the spelling.

6) "in vivo" should be italic.

A: Thanks, we have changed all "in vivo" to italic.

Reviewer #4

Q: In the manuscript, the authors constructed a reversible arylazo-conjugated fluorescent probe HDSF to detect cycling hypoxia. They explain the chemical mechanism of the reversibility by oxygen levels. They show the in vivo use of the reagent by drug-induced

hypoxia in the zebrafish embryo, in tumor xenograft in mice, and ischemia-reperfusion of the mouse limb. It is intriguing since the probe is reversible and can detect hypoxia in the tumor in mice. The authors claim HDSF is the first organic fluorescent probe capable of reversibly real time imaging of drug-induced and mechanical force induced hypoxia in vivo. However, there are some questions and concerns.

A: Thanks for your comments, we have prepared a point to point response below.

Q1: There are many data in the supplement (up to 19), including important information. Some data can be integrated into the main text. For example, Figure S9 with Figure 1, and Figure S14 with Figure 2 can be combined to make a clear contrast.

A: Thanks for the suggestion, we have chosen some data in SI and integrated into main text.

Q2: In Figure 6, it looks like the probe detects mild intramuscular hypoxia before the treatment in both legs. The tourniquet-induced ischemia in 15 min increases the hypoxic signal, suggesting the response to the gradient of O₂ level. But Figure 3 and Figure S17 show that the probe detects mild hypoxia as 10% O₂ in cells and does not show much difference between 10% and 0% O₂. How is it explained?

A: Images in Figure 6 were taken by PerkinElmer IVIS Lumina K Series III *in vivo* imaging system, and images in Figure 3 and Figure S17 were taken by confocal microscope Zeiss LSM710. Sensitivity of equipment may be different. The data in Figure 6 showed the gradual hypoxia process caused by tourniquet treatment. As the time extended, hypoxia condition became more severe and the activity of azoreductase increased, resulting in a gradual fluorescence enhancement. However, the data in Figure 3 and Figure S17 showed the final result of a fixed hypoxia condition (0.1% O₂) for 2 h without capturing the fluorescence turn-on process.

Q3: Figure 6 indicates the local injection of the probe does not expand and stays locally. Does systemic injection (intra-vascular) detect the whole ischemic limb during shutting blood flow? How long may it last in vivo?

A: Thanks for the comment. We have done additional experiments as the reviewer suggested. 50 μL HDSF solution of different concentration (20 μM, 100 μM, 200 μM) was injected via tail vein, and tourniquet-induced ischemia was created at different time point over 0-7 h post injection (bind for 25 min before imaging every time). However, no obvious fluorescence signal was detected in the ischemia limb (Fig R11). This data might be attributed to the concentration of HDSF located in limb by i.v. injection is not enough to create a noticeable fluorescent signal.

Fig. R11 Optical imaging of ischemia process in a living mouse after HDSF (20 μM, 50 μL)

was injected via tail vein.

Q4: Why were the nude mice used for ischemia-reperfusion limb study? Does the probe cause immunoreactivity?

A: Nude mice are used because they are more convenient for fluorescence *in vivo* imaging. Immunoreactivity was not considered in our experiment. According to the biosafety data, HDSF showed negligible toxicity *in vivo* at the experiment concentration.

Q5: There are 3 videos in the supplement, but no explanation in detail, except it looks like Doppler ultrasound images of the ischemia-reperfusion process of mouse limb.

A: A special word file has added to explain the 3 videos. Besides, details about the test is now available in the Method part and the note of supplementary Fig. 20.

Q6: Figure S13. Are DMPO only (a) and control without DMPO (b) in hypoxia? Does it mean hypoxia generates radicals, but hypoxic HDSF may not? In the text, "This data demonstrated that the single electron reduction product ...of HDSF was not stable enough..., suggesting a new mechanism different from previously reported arylazo-reduction based hypoxia probes". But later with Fig S17, they state, "fluorescence increment under hypoxia was a result of HDSF reduction." It is confusing.

A: Thanks for the comments. Previous reported arylazo-reduction based hypoxia probes undergo a process illustrated in Figure 1a, with a radical anion (**one electron reduction product**) as intermediate, which can be monitored by EPR assay. Thus EPR tests were taken to check whether radical compound exists during HDSF reduction in hypoxia just as previous reported probes do. The reaction solution was used for EPR test (supplementary Fig. 13a, Fig. R12a), and no obvious signal was found. Whether the radical is unstable enough to be captured directly? Then a radical capture agent DMPO was added to stabilize radicals. However, there was still no obvious radical signal (supplementary Fig. 13b, Fig. R12b). The EPR and HPLC-MS assay (Figure 2a, 2b, supplementary Fig. 14, Fig. R13, Fig. R14) indicated that HDSF was reduced through a new mechanism (probably **two electron reduction product**), which is different from previously reported arylazo-reduction based hypoxia probes.

Fig. R12 (a) EPR spectra of HDSF (1.5 mM) in PBS buffer (0.1 M, pH 7.4, 10% DMF, v/v) containing rat liver microsomes (RLM, 20 mg/mL) and NADPH (3 mM) incubated at 37°C in hypoxia condition (glove box) for 15 min. Ctrl: PBS containing RLM and NADPH. (b) Radical capture agent 5,5-Dimethyl-1-pyrroline-N-oxide (DMPO) was added in the

mentioned solution in the beginning or at the end of reaction respectively. Signal of pure DMPO was collected as a control.

Fig. R13 (a) ESI-MS spectra of probe solutions (20 μ M in PBS buffer) incubated with RLM (250 μ g/mL) and NADPH (100 μ M), (a) HDSF incubated in hypoxic condition. (b) HDSF solution exposed in air after incubated in hypoxic environment.

Fig. R14 HPLC spectra of HDSF solution (20 μ M in PBS buffer, pH 7.4) incubated with RLM (250 μ g/mL) and NADPH (100 μ M) at 37°C for 20 min (a) in air, (b) in hypoxic condition (glove box), (c) solution of (b) exposed in air for 30min. Elution solvent: mixture of methanol and water (v/v = 1/1) containing 0.1% TFA. Data collected at 680 nm.

Q7: Figure S17. Dicoumarol is a Vitamin K-relating reductase inhibitor. DPI is a flavoenzyme inhibitor that may quench superoxide radicals. Please adjust the interpretation. Does this suggest HDSF reacts to superoxide radicals?

A: Thanks for the suggestion, we have adjusted our interpretation. As illustrated in the additional selectivity study (supplementary Fig. 11b, Fig. R9), HDSF did not react to superoxide radicals.

Q8: A reversible two-photon luminescent probe has been reported to show cycling hypoxia in cells and zebrafish (Zhang P, 2015, PMID25890748), and various hypoxia

probes are clinically used to trace tumors (Mirabello, review 2018 PMID29527524). The novelty does not appear so strong as the authors claim. They should explain the uniqueness and advantage of using their probe.

A: Thanks for this comment. Indeed, noble metal complex based phosphorescent probes have been used for hypoxia monitoring, several great works have been reported including the paper (Zhang P, 2015, PMID25890748) mentioned by the reviewer. This type of probes utilized the O₂ sensitivity of the phosphorescence. However, one should consider that the noble metals are usually expensive, which will increase the economic cost of the final metal based photosensitizer. In addition, metal complexes are potentially toxic to biological samples, especially when irradiated with light they could generate ROS (¹O₂). In this regard, small organic molecule based probes for hypoxia imaging stand out as an attractive alternative. Although some organic small molecule based hypoxia probes were reported, most of them are irreversible. Therefore, considering the features of HDSF such as reversibility, mitochondria targeting, NIR emission and its application in real time *in vivo* monitoring ischemia-reperfusion process for the first time, we are confident that HDSF showed its uniqueness and advantages compare to the existing hypoxia probes.

Minor points:

Q9: Figure S16. "HIF-1a level in cell supernatant" must be in "cell lysate." HIF-1a is not secreted protein. Western blot does not indicate molecular size. It is hard to appreciate the HIF-1 induction by hypoxia compared to normoxia.

A: Thanks for the comment. The HIF-1a level assay was designed to evaluate the fluctuation of O₂ content in cells. Now a commercial hypoxia (irreversible) fluorescent probe Image-iTTM Green (Invitrogen) was used to verify the change of cellular O₂ content. Cells were co-stained with Hypoxia reagent Image-iTTM Green and HDSF in normoxia-hypoxia cycles. As shown in supplementary Fig. 16 (Fig. R8), no obvious fluorescence appeared in normoxia cells. After cells treated in AneroPack[®] (Mitsubishi Gas Chemical Company, Inc.), both green fluorescence (Hypoxia reagent Image-iTTM Green) and red fluorescence (HDSF) appeared, revealing a hypoxia environment in cells and HDSF was able to image it. When cells exposed to air and O₂ content increased, red fluorescence of HDSF dimmed, and green fluorescence of Hypoxia reagent Image-iTTM Green stayed still because of the irreversibility.

Q10: They should explain the acronym when it appears first in the text for general audiences. (e.g., NIR, HDSF, DFT, EST-MS, ESI). Also, Scheme 1 is hard to understand without much explanation.

A: Thanks for the comments. We have explained the acronym when it appears for the first time in the main text. For better understanding of Scheme 1, we have changed some of our interpretation. Scheme 1 calculated the free energies of some key intermediates and transition states, which provide information of the energy barriers for the irreversible N-N bond cleavage. As discussed in the text and illustrated in Figure 1a,b (Fig. R15) and TOC (Fig. R16), the energy barrier needed to overcome for the N-N bond cleavage of HDSF-HZ is too high, suggesting that it was not allowed in physiological temperature. This theoretical data supported that the CF₃-modification stabilized the two electron reduction

intermediate HDSF-HZ and prevented it from further irreversible reductive cleavage, which offered the possibility for reversible hypoxia sensing.

Fig. R15 (a) Normal azobenzene-derived fluorescent probes for hypoxia and their reductive decomposition of N=N bond by hypoxic microenvironment in living systems. (b) Reversible hypoxia sensing mechanism of probe HDSF.

Fig. R16 Diagram illustrating the energy barriers needed to overcome for the HDSF reduction.

Q11: The last part of the Methods should be about **Figure 6, not Figure 5.**

The references are in an odd format. It should list only one publication with each number.

A: Thanks for the comments. We have change the Methods part and the references format.

REVIEWERS' COMMENTS

Reviewer #1 (Remarks to the Author):

I suggest acceptance of the revised manuscript since all my concerned questions have been settled down.

Reviewer #2 (Remarks to the Author):

I think most my comments have been addressed sufficiently.

I would change one sentence in the manuscript;

With transgenic zebrafish embryos signed certain organs with fluorescent proteins

Using transgenic zebrafish where certain organs are fluorescently marked...

Freek van Eeden

Reviewer #3 (Remarks to the Author):

The authors have fully answered my comments and made appropriate modification in the reversion. Therefore, I recommend it be accepted for publication.

Reviewer #4 (Remarks to the Author):

The authors answered and explained well to my concerns

REVIEWERS' COMMENTS

Reviewer #1 (Remarks to the Author):

I suggest acceptance of the revised manuscript since all my concerned questions have been settled down.

A: Thanks for the valuable comments.

Reviewer #2 (Remarks to the Author):

I think most my comments have been addressed sufficiently.

I would change one sentence in the manuscript;

With transgenic zebrafish embryos signed certain organs with fluorescent proteins

Using transgenic zebrafish where certain organs are fluorescently marked...

Freek van Eeden

A: Thanks for the comments and suggestion, we have changed the sentence accordingly.

Reviewer #3 (Remarks to the Author):

The authors have fully answered my comments and made appropriate modification in the reversion. Therefore, I recommend it be accepted for publication.

A: Thanks for the valuable comments.

Reviewer #4 (Remarks to the Author):

The authors answered and explained well to my concerns.

A: Thanks for the valuable comments.